# CSEANet: Cross-Stage Enhanced Aggregation Network for Detecting Surface Bolt Defects in Railway Steel Truss Bridges

**DOI:** 10.3390/s25113500

**Published:** 2025-05-31

**Authors:** Yichao Chen, Yifan Sun, Ziheng Qin, Zhipeng Wang, Yixuan Geng

**Affiliations:** 1Beijing Jiaotong University, Beijing 100080, China; 18612010366@163.com (Y.C.); 22281196@bjtu.edu.cn (Z.Q.); 2CRRC Qingdao Sifang Co., Ltd., Qingdao 266111, China; m13626416757@163.com

**Keywords:** bolt defect detection, small object detection, UAV-based inspection, multi-scale feature fusion, railway safety

## Abstract

The accurate detection of surface bolt defects in railway steel truss bridges plays a vital role in maintaining structural integrity. Conventional manual inspection techniques require extensive labor and introduce subjective assessments, frequently yielding variable results across inspections. While UAV-based approaches have recently been developed, they still encounter significant technical obstacles, including small target recognition, background complexity, and computational limitations. To overcome these challenges, CSEANet is introduced—an improved YOLOv8-based framework tailored for bolt defect detection. Our approach introduces three innovations: (1) a sliding-window SAF preprocessing method that improves small target representation and reduces background noise, achieving a 0.404 *mAP* improvement compared with not using it; (2) a refined network architecture with BSBlock and MBConvBlock for efficient feature extraction with reduced redundancy; and (3) a novel BoltFusionFPN module to enhance multi-scale feature fusion. Experiments show that CSEANet achieves an *mAP*@50:95 of 0.952, confirming its suitability for UAV-based inspections in resource-constrained environments. This framework enables reliable, real-time bolt defect detection, supporting safer railway operations and infrastructure maintenance.

## 1. Introduction

The detection of bolt defects in railway steel truss bridges is crucial for ensuring structural safety and maintaining the stable operation of railway traffic. Railway authorities must inspect for bolt loosening, detachment, and corrosion in order to develop an appropriate maintenance plan. Currently, railway bolt inspection still relies mainly on manual checks and basic instruments. This approach is inefficient and subjective, making it difficult to ensure comprehensive and accurate assessments. Manual inspection suffers from limitations such as inspector fatigue, challenging working environments, and inadequate detection tools, which can lead to misjudgments and omissions. Additionally, the complexity of steel truss structures and the high-altitude nature of the work, combined with the limited range of inspection equipment, make manual methods dangerous, costly, and difficult to carry out. Therefore, it is essential to explore more efficient and automated inspection methods to enhance both inspection efficiency and accuracy.

In recent years, due to its flexibility, low cost, and remote control advantages, unmanned aerial vehicle (UAV) inspection technology has been applied in railway steel truss bridge inspection. UAVs can quickly fly over a large area to transcend the geographical location of bridges and take high-resolution image data, which can reduce the amount of work for manual inspections and improve data collection efficiency. Railway inspection UAV technology has made remarkable achievements in a variety of railway inspection scenarios [1,2,3,4,5,6]. Although there are obvious advantages of UAVs in data acquisition, the following challenges still exist in the task of detecting bolt defects in railway steel truss bridges: small target detection, interference from complex backgrounds and limited computational resources.

In the images captured by UAVs, bolts are extremely small targets, often occupying only a few pixels even in high-resolution imagery. This presents a major challenge for traditional object detection algorithms, which tend to lose critical features of small objects during the feature extraction process, resulting in decreased detection accuracy. Furthermore, the complex background of railway steel truss bridges—characterized by overlapping components, lattice structures, and variable lighting—can easily obscure small targets and introduce noise, leading to frequent false positives and missed detections. High-speed UAV movement can further amplify motion blur and lighting inconsistency, degrading the quality of image data. While anchor-based detection methods offer strong localization for larger objects, their performance significantly declines for small-scale objects due to limitations in anchor design and receptive field coverage. Although prior research has shown that region-based methods like Faster R-CNN can achieve high precision in detecting blurry and small-scale defects in GPS-denied environments [7], these models often require considerable computational resources, making them impractical for lightweight UAV platforms. Consequently, improving the accuracy and efficiency of small object detection has become an urgent need in UAV-based railway inspection scenarios.

Other than the challenges in small target detection, UAV-based inspection also faces practical limitations from both hardware and operational perspectives. On the one hand, UAVs are typically equipped with resource-constrained onboard devices, which makes it difficult to deploy high-precision detection algorithms that demand significant computational power. On the other hand, the reliance on licensed pilots increases operational costs and limits scalability, especially in complex or repetitive inspection tasks [8]. These factors have driven the development of autonomous UAV systems. Furthermore, to enhance flight safety in environments with unpredictable obstacles, recent studies have incorporated deep learning-based obstacle avoidance into autonomous UAV platforms [9]. Therefore, detection algorithms must not only achieve high accuracy but also be lightweight, real-time, and adaptable to autonomous UAV systems. Compared with traditional two-stage detection algorithms, traditional two-stage detection algorithms such as R-CNN and Fast R-CNN [10,11,12] have high detection accuracy but large computational complexity, which is unacceptable in UAV inspection application. While one-stage detection algorithms such as YOLO and SSD [13,14] can quickly predict the location and category of objects in an end-to-end manner and have fast inference compared with two-stage algorithms, there is still space for accuracy improvement for small target detection.

To increase the accuracy and efficiency of small target detection, researchers have proposed various improvement strategies, such as data augmentation [15,16], multi-scale feature learning [17], and Feature Pyramid Networks (FPNs) [18,19,20]. Furthermore, the introduction of attention mechanisms, such as channel attention (SE), spatial attention (CBAM), and coordinate attention (CA), has significantly enhanced the model’s ability to perceive key regions, improving both the accuracy and robustness of object detection [21,22,23].

In addition to boosting the detection accuracy, researchers also attempt to improve the computational efficiency by designing some lightweight methods. For instance, DSC [24] reduces the computational cost while maintaining certain feature extraction ability. Furthermore, GSConv [25] and Gold-YOLO [26] also optimize the feature pyramid structure to balance the multi-scale feature fusion efficiency and computational complexity by improving the model lightweighting. In recent years, anchor-free detection strategies have also gained significant attention [27,28], as they eliminate dependency on predefined anchor boxes, enhancing the model’s adaptability to small target detection and demonstrating potential applications in UAV-based inspection tasks. The integration of these optimization strategies provides new research directions for intelligent bolt defect detection in railway steel truss bridges while also advancing the application of computer vision in infrastructure inspection.

The YOLO series [29,30], particularly the YOLOv8 algorithm released in 2023 [31], has demonstrated high accuracy and speed in object detection tasks. However, YOLOv8 still exhibits certain limitations when dealing with complex backgrounds and small target detection, especially in scenarios where objects vary significantly in size or where background interference is substantial. In such cases, its performance may be inferior to some specialized small-object detection algorithms. Consequently, existing detection methods struggle to fully meet the requirements for bolt defect detection in railway steel truss bridges.

While previous studies have addressed some of these issues through targeted architectural or loss function improvements, they often focus on isolated components and fail to offer an integrated solution that simultaneously meets the demands of small target accuracy, real-time performance, and UAV-deployable efficiency. Moreover, most YOLO-based improvements do not explicitly consider the difficulty of cross-stage feature interaction in complex industrial scenes, nor do they account for deployment constraints of onboard UAV systems.

To bridge this gap, our study aims to move beyond isolated module enhancements and propose a system-level framework that integrates novel data preprocessing (SAF), optimized backbone design (BSBlock and MBConvBlock), and a refined neck structure (BoltFusionFPN) under a unified detection pipeline. This approach is designed to support real-world bolt defect detection under the constraints of high-resolution UAV imagery, limited onboard computation, and dense background interference. We believe this integration not only advances the technical performance but also provides a more generalizable and practical paradigm for future research in UAV-based infrastructure inspection.

The main contributions of this paper can be summarized as follows:To address the challenge of small target detection for bolts in large images captured by UAVs, we propose a sliding-window-based SAF (slice-assisted fine-tuning) preprocessing method. During the training phase, images are divided into overlapping patches, and these patches are processed and then combined to form a complete image. This reconstruction helps improve the feature representation of small objects by focusing the model’s attention on smaller, localized areas within the image, rather than on irrelevant background regions. In the inference phase, the patches are stitched back together without resizing, ensuring that small objects are not distorted by resampling. This method also helps eliminate redundant background information, making the model more efficient by concentrating on the relevant regions of the image, such as bolt defects. Additionally, by combining cross-stage data augmentation strategies, we expand the original set of 50 images to 1115 images, resulting in an improvement in the model *mAP* to 0.818 (a 97.1% increase compared to the non-sliced version).To balance lightweight design with high accuracy, we first replace the deep C2f module in the backbone network with BSBlock, a deep feature extraction module based on PConv and PWConv. Simultaneously, we introduce MBConvBlock, a shallow feature extraction module integrating MBConv and CBAM, to replace the shallow layers of the network. This optimization enhances texture feature extraction while reducing parameter redundancy.To address the issue of feature loss caused by PANet’s reliance on intermediate layer information transfer, we design a novel neck network, BoltFusionFPN. This network introduces the collection–distribution (GD) mechanism of Gold-YOLO, enabling the direct alignment of non-adjacent layer features. Combined with the channel shuffling operation of GSConv, it facilitates the dynamic fusion of multi-scale features.

The following part of this paper is organized as follows: Section 2 reviews related work, particularly focusing on algorithms improved from YOLO for small target detection. Section 3 provides details on the network structure of the CSEANet model and the related modules. Section 4 presents experimental results and performance evaluations. Finally, conclusions and future work are drawn in Section 5.

## 2. Related Works

Before the adoption of deep learning in bolt defect detection, several vision-based methods were developed to identify loosened bolts. For instance, one approach utilized the Hough transform combined with a support vector machine to extract geometric features—such as bolt head length and orientation—from smartphone images, achieving reliable quasi-real-time classification performance [32]. Another notable method applied the Viola–Jones algorithm to automatically localize bolts and calculate damage-sensitive features, which were subsequently fed into an SVM classifier to distinguish loosened from tight bolts [33]. These traditional methods demonstrated the feasibility of bolt detection using classical computer vision techniques. However, their effectiveness often depends heavily on controlled lighting, fixed viewpoints, and simple backgrounds, which limit their applicability in complex UAV-based inspection scenarios. Consequently, recent research has increasingly focused on deep learning models that offer greater robustness and automation under diverse conditions.

The first application of convolutional neural networks (CNNs) to bolt damage detection was presented in an autonomous inspection framework using region-based deep learning to identify multiple types of structural defects [34]. More broadly, deep-learning-based structural health monitoring has gained increasing attention, with key works addressing general SHM frameworks [35] and concrete crack detection using CNNs [36]. In recent years, the YOLO (You Only Look Once) series of algorithms have been extensively studied for small object detection tasks. Due to challenges such as insufficient feature representation, severe background interference, and complex scale variations, researchers have made improvements to YOLO in areas such as feature extraction, feature fusion, loss function optimization, and lightweight design to enhance detection accuracy and efficiency.

In terms of feature extraction, researchers have enhanced the representation capability of small objects by optimizing backbone network structures and incorporating efficient attention mechanisms. For instance, Li and Shen [37] applied super-resolution reconstruction techniques to make the features of low-resolution small objects clearer, thus enhancing detection accuracy. Su and Qin [38] introduced multi-level feature integrators and perceptual-enhanced convolution modules based on YOLOv8, making the feature representation of small objects more robust. Hui et al. [39] proposed a structural improvement based on Swin Transformer, further strengthening the feature extraction ability for small objects. Additionally, Zeng et al. [40] improved the feature aggregation capability of detection networks through the PC-C2f and ASD-FPN structures, significantly enhancing small object recognition.

In terms of feature fusion, optimizing multi-scale feature extraction and fusion strategies is key to improving small object detection capabilities. Some studies have enhanced the transmission and utilization efficiency of features across different scales by improving the feature pyramid structure (FPN) and adaptive fusion strategies. For example, Bi and Li [41] optimized the feature fusion structure of YOLOv8, improving detection performance for objects at different scales. Wang and Zhou [42] enhanced the model’s ability to capture small object features by introducing RepGhost and Normalized Attention Modules (NAMs). Li and Zheng [43] designed the RepNCSPELAN4 and C2FCBAM modules, strengthening the interaction between features at different levels, thus balancing detection accuracy and computational efficiency. Ma et al. [44] proposed DLW-YOLO, combining the SPPELAN and VoV-GSCSP frameworks, which effectively improved small object detection performance in complex environments.

In terms of loss function optimization, researchers have focused on alleviating the regression error bias caused by small object sizes during the training process. For example, Xu et al. [45] introduced a balanced Focal Loss based on YOLO, which made the model pay more attention to small objects and reduced false positives and false negatives in detection. Xu and Xiong [46] proposed EMA-YOLO, which improved the object localization loss and enhanced the detection stability of small objects, particularly in complex backgrounds. Zhong and Zhang [47] optimized YOLO’s loss function design by combining multi-order gated aggregation modules, further enhancing detection accuracy and robustness.

In terms of lightweight design, to reduce computational overhead and improve real-time performance, several studies have employed model compression, parameter optimization, and efficient computational units. For example, Betti and Tucci [48] proposed YOLO-S, which uses skip connections and lightweight structures, making the model suitable for resource-constrained scenarios while maintaining high detection accuracy. Xu [49] optimized YOLOv3 by combining auxiliary networks and attention mechanisms, reducing computational complexity while improving detection accuracy. Zhang and Meng [50] introduced MBAB-YOLO, which strikes a good balance between high-precision detection and real-time performance by using adaptive multi-receptive field focusing and hybrid attention modules. Huangfu et al. [51] proposed Ghost-YOLO v8, which integrates GhostConv and the SE mechanism, enabling the model to effectively capture small object features while maintaining efficient computational capabilities.

Additionally, some studies have further integrated specific structural optimizations to enhance detection performance. Hui [52] enacted SEB-YOLO using SPD-Conv and Bi-FPN structure to enhance semantic information extraction ability and enhance the accuracy of small object detection. Shin et al. [53] designed DCEF2-YOLO. Deformable convolutions and an efficient feature fusion structure are introduced into YOLO, and the model’s adaptability to small objects is enhanced. Xu and Dong [54] added an attention mechanism to a YOLO network. The method greatly improved the detection effect of small anomalous objects in complex backgrounds. Xia [55] proposed TTD-YOLO, which improves the network structure to provide more stable performance in multi-scale object detection. Wu [56] proposed SAW-YOLO, which further enhances detection performance by combining an optimized feature fusion module.

In summary, YOLO-based improvements have made significant progress in small object detection tasks. Researchers’ explorations in feature extraction, feature fusion, loss optimization, and lightweight design have continuously enhanced detection accuracy and computational efficiency. However, in the specific industrial scenario of detecting bolt defects on the surface of railway steel truss bridges, existing methods still face challenges such as insufficient cross-stage feature aggregation and the difficulty of balancing detection accuracy with real-time performance under UAV hardware constraints.

Most of these approaches focus on isolated module-level improvements and lack a unified framework tailored for UAV-based bolt inspection. To address these issues, this study proposes a Cross-Stage Enhanced Aggregation Network (CSEANet), which systematically integrates SAF-based preprocessing, lightweight feature extraction modules, and an optimized neck structure. This comprehensive design enhances feature representation and fusion, thereby improving detection accuracy while ensuring real-time performance for deployment in practical UAV inspection tasks.

## 3. Methodology

### 3.1. Framework Overview

We propose an improved object detection network, CSEANet, to enhance the accuracy and robustness of surface bolt defect detection in railway steel truss bridges. Traditional detection methods, including YOLOv8, often struggle with small object representation, ineffective feature fusion, and high computational cost.

To address these issues, CSEANet builds upon YOLOv8 with targeted structural improvements. It enhances the backbone and neck networks through the integration of MBConvBlock and BSBlock for better feature extraction and replaces PANet with the BoltFusionFPN module for more effective multi-scale fusion. Additionally, the SAF (slice-assisted fine-tuning) preprocessing strategy improves small object visibility and reduces background noise.

The standard YOLOv8 pipeline is shown in Figure 1 for comparison, while the overall architecture of our proposed CSEANet is illustrated in Figure 2. Compared with YOLOv8, CSEANet achieves better detection performance for small objects while maintaining high efficiency, making it well-suited for UAV-based bridge inspection tasks.

### 3.2. SAF Slicing Preprocessing

The primary challenge in surface bolt defect detection on railway steel truss bridges is that the features of small objects are easily obscured by background noise in large images. As illustrated in Figure 3, the pixel area of a bolt constitutes only 0.10% of the entire image, representing a typical small object detection scenario. Directly inputting the original high-resolution image into a detection network, such as YOLOv8, often leads to the loss of small object features due to excessive downsampling in deeper layers. On the other hand, shallow networks struggle to extract deeper semantic information, making it difficult to achieve a balance between detecting both large and small objects.

To address this challenge, this study proposes the SAF (slice-assisted fine-tuning) strategy, which extracts local patches from the original image and adjusts them to a larger aspect ratio during the fine-tuning process. This approach combines two key mechanisms, local feature enhancement and global context retention, significantly improving the model’s ability to detect small objects. In our implementation, SAF slicing is applied consistently in both the training and inference phases. During training, each image is divided into overlapping patches of 640 × 640 pixels, and slicing is performed per image, not per category. During inference, the input image is similarly sliced into 666 × 500 patches, and all predictions are projected back to the full image, followed by a global non-maximum suppression (NMS) step to remove duplicate detections in overlapping regions.

The original image is shown as $I_{1}^{F},I_{2}^{F},\ldots ,I_{j}^{F}$, where each image is divided into overlapping patches, $P_{1}^{F},P_{2}^{F},\ldots ,P_{k}^{F}$. Then, the size of all patches is adjusted, and the enhanced image is obtained as $I_{1}^{\prime },I_{2}^{\prime },\ldots ,I_{k}^{\prime }$. In both the training and inference phases, different SAF slicing preprocessing methods are applied to the dataset. Two images are randomly selected from the dataset as examples, as shown in Figure 4, Figure 5, Figure 6 and Figure 7. In the training process, we cut each original image into 64 overlapping patches and rebuild the dataset by resampling each patch into a 640 × 640-sized patch, while in the inference process, we cut each original image into overlapping patches of size 666 × 500 and no resampling is performed. After the inference is complete, the patches are stitched back together.

### 3.3. Shallow Feature Extraction Module MBConvBlock Based on MBConv and CBAM

In object detection tasks, the backbone network plays a crucial role in multi-level feature extraction, with the quality of shallow features directly influencing the detection capability of small objects. However, the C2f (Cross-Stage Partial Fusion) module used in YOLOv8 primarily focuses on optimizing gradient flow and lacks a dedicated mechanism to enhance small-object features. Its bottleneck structure relies on a large receptive field for feature aggregation, which often leads to the loss of critical information in small object detection tasks. Additionally, C2f uses channel decomposition and fusion for information transfer but does not integrate a dedicated attention mechanism, preventing the model from effectively focusing on small object regions during the feature extraction process. This uniform approach to image information processing works relatively well for large object detection but struggles to precisely capture defect features in small object tasks, such as surface bolt defect detection in railway steel truss bridges, ultimately limiting detection accuracy.

To address the limitations of C2f in small object detection, this study introduces MBConvBlock in the first two C2f module positions of the backbone network. This replacement aims to enhance shallow feature extraction and improve the model’s sensitivity to small objects. The MBConv structure was originally derived from MobileNetV3 and later optimized in EfficientNet. It incorporates two key features: depthwise separable convolution and inverted residuals. These characteristics reduce computational costs while enhancing feature representation capabilities. However, directly applying the standard MBConv still presents certain limitations. To better meet the demands of small object detection, this study introduces targeted improvements. These include refining the attention mechanism and adjusting the activation functions.

In terms of attention mechanisms, this study introduces CBAM (convolutional block attention module) to replace the SE (Squeeze-and-Excitation) mechanism in the original MBConv. While SE performs channel-wise attention computation, it fails to capture spatial information. In contrast, CBAM combines both channel and spatial attention, enabling the network to focus more precisely on critical target regions while reducing attention to irrelevant background areas, thereby enhancing small object detection performance.

During the channel attention computation in CBAM, the input features are first processed through Global Average Pooling (GAP) and Max Pooling to extract channel information. The pooled features are then passed through two fully connected layers. Finally, the channel weights are generated using a Sigmoid activation function, as shown in Equation (Equation 1). This approach allows the model to more effectively focus on relevant channels and suppress less important ones, further improving detection performance, especially for small objects.(1)$$F_{c}=\sigma \left(W_{2}\delta \left(W_{1}\mathrm{GAP}\left(X\right)\right)\right)$$
where $W_{1}$ and $W_{2}$ are fully connected layer parameters, δ is the ReLU activation function, and σ is the Sigmoid normalization function. The feature enhanced by the channel attention module then enters the spatial attention module, where a 7×7 convolution is used to compute the spatial attention map. This allows the model to more precisely adjust the feature weights of small target regions, thereby improving detection accuracy.

Furthermore, to improve computational efficiency and enhance the stability of the model, this study replaces the Swish activation function in the MBConv structure with ReLU, whose specific formula is shown in Equation (Equation 2). While Swish has a higher computational complexity and can improve feature representation, it is not suitable for lightweight deployment scenarios. ReLU6, by limiting the range of activation values, helps reduce the risk of numerical overflow and improves computational efficiency. Its expression is as follows:(2)ReLU6(x)=min(max(0,x),6)

This adjustment not only reduces the computational load but also optimizes training stability, allowing the model to maintain stable performance even in environments with low computational resources. By using ReLU6, the model can effectively balance efficiency and accuracy, ensuring that it operates reliably under resource-constrained conditions without compromising its detection capability.

The overall process of the MBConvBlock is shown in Figure 5. The input feature $X\in R^{H\times W\times C}$ first passes through a 1×1 convolution layer for channel expansion, increasing the number of channels from C to αC, where α is typically set to 4, thereby enhancing the feature representation capability. Next, depthwise separable convolution is used for feature extraction. Unlike standard convolution, this operation splits the computation into depthwise convolution (which operates on individual channels) and pointwise convolution (which fuses channel information). This approach significantly reduces computational cost while maintaining the feature representation ability. Additionally, to better capture small object features at different scales, the study employs 3×3 or 5×5 variable convolution kernels in the depthwise convolution, allowing the network to handle small objects of various sizes more effectively.

After feature extraction, the CBAM mechanism further optimizes the feature representation, allowing the network to dynamically adjust feature weights and focus more on the bolt defect areas, rather than distributing attention uniformly across the entire image. Finally, a 1×1 convolution layer is used for channel compression, reducing the number of channels from αC back to the original dimension C, which helps decrease the computational cost. A dropout layer is also introduced at this stage to reduce the risk of overfitting and improve the model’s generalization ability. Overall, the integration of the MBConvBlock and CBAM structure enhances CSEANet’s ability to extract local features for small object detection tasks while reducing computational resource waste and improving overall detection performance.

### 3.4. Deep Feature Extraction Module BSBlock Based on PConv and PWConv

In object detection, above the deep layer of the backbone network, the deep network focuses on learning more global contextual information. Compared with shallow networks, deep networks learn more global information to enhance the discriminative ability of the model. However, more parameters will increase the training expense in deep networks. Furthermore, the problem of vanishing or exploding gradients may also appear due to the high computation cost of deep networks. Specifically, in the YOLOv8 backbone network, C2f serves as the core component for deep feature extraction. Although it has advantages in optimizing gradient flow, it faces two main issues: excessive computational redundancy and limited effectiveness in feature extraction.

The C2f module presents a significant computational burden. While its Bottleneck structure helps reduce channel redundancy, it still relies on large feature maps for processing, leading to a high number of parameters and floating-point operations (FLOPs). In high-resolution image tasks, such as bolt defect detection in railway steel truss bridges, the deep feature extraction stage requires handling extensive high-dimensional data. This heavy computational load greatly increases inference time, making it challenging for the model to maintain real-time performance. Moreover, the C2f structure prioritizes gradient propagation but does not fully leverage computational resources for effective feature selection. As a result, redundant information flows through the network, making it less efficient at capturing crucial target details. This limitation is especially problematic in small object detection, where weak deep feature representation can cause small targets to be overlooked, ultimately reducing detection accuracy.

To overcome these challenges, this study introduces the BSBlock (Bottleneck-Shuffle Block) into the deeper layers of the YOLOv8 backbone network as a replacement for the C2f module. This enhancement improves deep feature representation while significantly reducing computational overhead.

The structure of BSBlock is illustrated in Figure 6. First, the input feature $X\in R^{H\times W\times C}$ undergoes processing through partial convolution (*PConv*), which reduces computational overhead while emphasizing key target features. This allows computational resources to focus on the critical regions of bolt defects. The computation of partial convolution can be expressed as shown in Equation (Equation 3):(3)Y=PConv(X)=W⊙X+b
where *W* represents the weight matrix, ⊙ denotes element-wise multiplication, and *b* is the bias term. Compared to standard convolution, *PConv* performs computations only on valid regions, allowing computational resources to focus on the critical areas of bolt defects while reducing unnecessary calculations.

To enhance the interaction and fusion of channel information, BSBlock employs two Pointwise Convolution (PWConv) layers, effectively utilizing the information from different feature channels to improve the completeness of feature representation. Additionally, to maintain consistency of feature information in deep networks and mitigate the vanishing gradient problem, BSBlock incorporates residual connections.

In the CSEANet architecture, BSBlock replaces the third and fourth C2f modules of the YOLOv8 backbone. This adjustment not only reduces the model’s parameter count and computational complexity, contributing to its lightweight design, but also strengthens the robustness of deep feature extraction. As a result, CSEANet can more accurately preserve critical small-object information in bolt defect detection tasks for railway steel truss bridges without being affected by background noise or computational redundancy.

### 3.5. The Neck Feature Pyramid Based on GSConv and Gold-YOLO: BoltFusionFPN

In object detection tasks, the neck network is responsible for fusing multi-level features from the backbone network at different scales to ensure the effective combination of deep semantic information and shallow detail information. However, in the original YOLOv8 structure, the PANet (Path Aggregation Network) is used as the neck network, primarily performing multi-scale feature fusion through lateral and context paths. While PANet demonstrates good performance in object detection tasks, its hierarchical information fusion strategy still has certain limitations when handling small object detection tasks.

As shown in the diagram, PANet employs a hierarchical information interaction strategy, where low-level features need to be passed up to higher levels, while high-level features gradually flow back to lower levels to achieve fusion of global and local information. However, this stepwise feature propagation mechanism leads to two main issues: First, non-adjacent layers find it difficult to directly interact, and critical information for small objects may be weakened or even lost during the stepwise transmission process. Second, this mechanism increases computational overhead, as additional feature transmission paths introduce information redundancy, negatively impacting inference efficiency. This is particularly evident in high-resolution image tasks, where the computational burden is further amplified. Specifically, in the task of detecting bolt defects in railway steel truss bridges, the detailed features of small objects are often distributed in the shallow layers, and PANet fails to efficiently align these critical pieces of information, limiting the detection accuracy for small objects.

To address the above issues, this paper reconstructs the neck network based on Gold-YOLO, which, through its gather-and-distribute (GD) mechanism, directly aligns features across non-adjacent layers. Building upon this, the paper adopts a multi-scale feature fusion approach, making feature integration more diverse and effectively mitigating the issue of small object features being overwhelmed in very deep networks. Following this, the GSConv module is placed after the deep feature extraction module in the backbone network, reducing the number of parameters while maintaining or improving accuracy and lowering computational costs. BoltFusionFPN, by replacing the neck network in YOLOv8, expands the model’s learning capacity and absorbs feature information from different network layers, significantly enhancing the model’s learning and generalization abilities. The following sections introduce the improved modules in detail.

(1) Gold-YOLO Mechanism

The Gold-YOLO mechanism consists of two parts: Low-GD (Low-Level Feature Collection) and High-GD (High-Level Feature Collection). Its overall architecture is shown in Figure 2. In the Low-GD section, Gold-YOLO performs feature scale alignment and channel information fusion through the Low-Level Feature Alignment Module (Low-FAM) and Low-Level Information Fusion Module (Low-IFM). This ensures that features from different layers are fully fused before entering the higher-level stages. The feature alignment process is mathematically formulated in Equation (Equation 4):(4)$$F_{\mathrm{align}}=\mathrm{Concat}\left(\mathrm{AvgPool}\left(B2\right),\mathrm{AvgPool}\left(B3\right),B4,\mathrm{UpSample}\left(B5\right)\right)$$
where B2,B3,B4,B5 represent features extracted from different layers of the Backbone. The AvgPool and UpSample operations are used to adjust the feature scales, ensuring that all feature maps are aligned to the same size. Subsequently, the fused features are processed through the Low−LevelInformationFusionModule(Low−IFM) to further optimize feature representation, enabling sufficient interaction between different channels. The features are then enhanced using the RepBlock module. Finally, a Split operation divides the features into two subsets, which are utilized for subsequent high-level information distribution, as illustrated in Figure 3.

In the High-Level Feature Collection (High-GD) stage, Gold-YOLO employs the Transformer mechanism for global relationship modeling, allowing high-level features to directly integrate key low-level information without relying on intermediate-layer feature propagation. Traditional PANet suffers from significant computational redundancy during the feature propagation process. Gold-YOLO optimizes high-level feature representation through a self-attention mechanism, enabling the efficient interaction of information across different scales. The computation is as follows.

In the High-Level Feature Collection (High-GD) stage, Gold-YOLO employs the Transformer mechanism for global relationship modeling, allowing high-level features to directly integrate key low-level information without relying on intermediate-layer propagation. As shown in Equation (Equation 5), the fused representation $F_{\mathrm{fuse}}$ is generated by applying a Transformer encoder to the aligned low-level features $F_{\mathrm{align}}$:(5)$$F_{\mathrm{fuse}}=\mathrm{Transformer}\left(F_{\mathrm{align}}\right)$$

The Transformer, through its global attention mechanism, allows high-level features to effectively complement low-level features without the need for gradual feedback. This mechanism enhances the model’s ability to perceive small target features while improving computational efficiency.

Furthermore, to further optimize the allocation of feature information, Gold-YOLO uses the Inject module for dynamic feature distribution, ensuring that high-level semantic features are accurately fed back to the shallow layers while also ensuring that shallow local features are effectively transmitted to the detection head after fusion. The Inject mechanism employs Local–Global Fusion, enabling the adaptive integration of the local features at the current scale, Xlocal, and the cross-scale global features, Xglobal.

(2) GSConv

To further alleviate the issue of excessive parameter size in the deeper layers of the original YOLOv8 feature extraction network, this paper places GSConv after the BSBlock module to bridge the deeper layers of the backbone network with the shallower layers of the neck network.

The network structure of GSConv is shown in Figure 7. It adopts a two-stage computational structure: first, standard convolution (Conv) is applied to the input features for initial downsampling, enhancing the feature representation capability; then, depthwise separable convolution (DWConv) is used to extract features from the convolved results, reducing computational complexity while preserving key spatial information. These two computational results are concatenated along the channel dimension and undergo a channel shuffle operation to enhance the interaction of information between channels, such that the final output has twice the number of channels as the concatenated input feature maps.

## 4. Experiment

### 4.1. Datasets and Parameter Setting

To evaluate the effectiveness and overall performance of the proposed architecture, this study constructs a dataset focused on the surface bolts of railway steel truss bridges. The dataset consists of raw images collected and preprocessed using the SAF slicing method. The image acquisition process is carried out by a DJI M30 UAV, which is equipped with a stabilized gimbal capable of four-degree-of-freedom rotation. The camera supports 16× hybrid optical zoom, ensuring stable and clear imaging of ground infrastructure from altitudes of up to 100 m, which meets the precise data collection requirements for railway steel truss bridge inspections.

The original dataset comprises 53 images, with 46 images designated for training and 7 images for validation. After SAF preprocessing, the dataset expands to a total of 1115 images, including 891 training images, 112 validation images, and 112 test images. As shown in Table 1 after SAF preprocessing, all images were resized to 640 × 640 pixels for consistency and effective training. The dataset composition is summarized below.

The input image dimensions vary depending on the model configuration. The dataset is labeled into three mutually exclusive categories: boltCorrosion (bolts showing visible rust, pitting, or surface material degradation), boltMissing (positions where the bolt is completely absent or visibly disconnected), and boltNormal (bolts with no apparent structural damage or corrosion).

In total, the dataset contains 10,221 instances of boltNormal, 3956 instances of boltCorrosion, and 34 instances of boltMissing, revealing a class imbalance that may impact detection performance on rare defect types. All annotations were manually verified by two certified railway bridge inspection experts to ensure consistency and reliability in the labeling process.

The experiments are conducted on a system running Ubuntu 18.04 with Python 3.8, PyTorch 1.7.1, CUDA 11.1, and cuDNN 8.0.4, utilizing an NVIDIA GeForce RTX 4090 GPU. More settings are shown in Table 2.

### 4.2. Evaluation Metric

In object detection, several metrics are used to evaluate the accuracy of a model. True Positives (*TP*s) are the correctly detected objects, i.e., the objects that are correctly identified by the model. False Positives (*FP*s) are the incorrectly detected objects, i.e., the objects that the model incorrectly identifies as present. False Negatives (*FN*s) are the objects that the model fails to detect, i.e., the objects that are present but not identified by the model.

Precision (*P*) is the ratio of *TP*s to the sum of *TP*s and *FP*s. It indicates the accuracy of the positive predictions made by the model:$$P=\frac{TP}{TP+FP}$$

Recall (*R*) is the ratio of *TP*s to the sum of *TP*s and *FN*s. It measures the model’s ability to identify all relevant objects in the dataset:$$R=\frac{TP}{TP+FN}$$

Additionally, Mean Average Precision (*mAP*) is calculated by taking the average of the precision values at different recall levels for each class and then averaging the results across all classes. *mAP* combines both precision and recall, providing an overall measure of the model’s performance.$$mAP=\frac{1}{N}\sum_{i=1}^{N}AP_{i}$$
where $AP_{i}$ is the Average Precision for class *i* and *N* is the number of object classes in the dataset.

### 4.3. Comparison Experiment on SAF Slicing Preprocessing

Before conducting the ablation experiments, we first evaluated the impact of SAF preprocessing by training the dataset before and after SAF slicing using the original YOLOv8s network and compared the performance differences. The evaluation metric used in this experiment is mean Average Precision (*mAP*), and the results are shown in Table 3.

As observed in Table 3, without modifying the network structure or changing the overall pixel composition of the dataset, simply applying SAF slicing during training led to a substantial performance gain. This improvement can be attributed to the exclusion of irrelevant image regions: after slicing, patches without labeled targets are no longer passed through the network, effectively filtering out background noise. This reduces the burden on convolutional layers and enhances the network’s focus on meaningful features.

It is also important to note that SAF slicing is applied not only during training but also in the inference phase. During inference, each image is sliced into overlapping patches of 666 × 500 pixels, and detection results from all patches are concatenated back to the original image coordinates.

Next, an inference comparison of the algorithm’s schematic diagram is conducted. At this stage, the training weights obtained after SAF slicing preprocessing are uniformly used. However, during inference, two cases are handled separately: one without SAF slicing preprocessing and the other with SAF slicing preprocessing. The results are shown in Figure 8.

The final inference example images provide an intuitive comparison of the differences before and after applying SAF slicing preprocessing during the inference stage. In the results without SAF preprocessing, as shown on the left side of Figure 8, the number of detected bolt targets is relatively low compared to the right side of Figure 8. Additionally, on the left side of Figure 8, the boltNormal” label in the lower right part is misclassified as boltCorrosion”. In contrast, detection results on the right side of Figure 8 show that the misclassification is resolved.

From the above experiment, it can be concluded that SAF slicing preprocessing effectively enhances both the detection precision and the accuracy of the algorithm, whether applied during the training or inference stage.

### 4.4. Ablation Study

After demonstrating that SAF slicing preprocessing effectively improves model performance on datasets with large image sizes and a high proportion of small objects, the ablation experiments on CSEANet were conducted using datasets that had been preprocessed with SAF slicing. For the inference example results, the images are first applied with SAF preprocessing, and concatenated back form the original image after the model inference. The results of the ablation studies are shown in Table 4, Table 5, Table 6 and Table 7.

The ablation experimental results indicate that, on average, across all labels, MBConvBlock significantly improves *mAP*, suggesting that, overall, the key to performance enhancement lies in the feature extraction capability of the shallow backbone network. When MBConvBlock + BSBlock were applied to modify the backbone network, or when only BoltFusionFPN was applied to modify the neck network, the overall *mAP* did not exceed 0.9. However, when both MBConvBlock and BSBlock were combined, the *mAP* reached 0.952, proving that both feature extraction capabilities and multi-level feature fusion were significantly improved.

For bolt corrosion detection, adding MBConvBlock, BSBlock, and BoltFusionFPN modules individually all improved *mAP*, but the combination of MBConvBlock and BSBlock resulted in a higher *mAP* than adding only BoltFusionFPN. This indicates that BoltFusionFPN still suffers from some information loss. For bolt missing detection, a significant *mAP* improvement was observed when MBConvBlock was added alone, indicating that missing bolt targets particularly benefit from enhanced shallow feature attention. Finally, for bolt normal detection, when all three modules were added, the *mAP* reached 0.945. Although this is slightly below the best result, it still represents a 2.9% improvement over the original YOLOv8, which demonstrates a significant enhancement in performance.

### 4.5. Comparative Experiment

To evaluate the performance of the proposed algorithm, we compared it with several mainstream object detection methods. Faster R-CNN [57] generates candidate regions through a Region Proposal Network (RPN), followed by a fully connected network for object classification and bounding box regression. It is well-suited for complex scenes. Cascade R-CNN [58] builds upon Faster R-CNN by utilizing a multi-stage detection strategy to refine detection results, particularly excelling in multi-scale and small object detection. RetinaNet [59] introduces the Focal Loss to address the class imbalance problem, demonstrating strong performance in highly imbalanced detection tasks. CenterNet [60] simplifies the detection process by regressing the center points and object sizes, enabling efficient target localization and classification, especially in real-time detection scenarios.

These methods employ different strategies to improve detection accuracy and speed, each with its own advantages and applicable scenarios. Comparing them with our proposed algorithm provides a comprehensive assessment of its performance in complex object detection tasks, and the results are shown in Table 8.

In this experiment, CSEANet demonstrated a significant improvement over other mainstream object detection algorithms, particularly in terms of accuracy. While traditional two-stage methods, such as Faster R-CNN, have long been recognized for their exceptional precision in small object detection tasks in complex scenarios, the single-stage model CSEANet, based on an improved YOLOv8, achieved a breakthrough in accuracy. This highlights the considerable potential of single-stage models in enhancing detection performance, particularly in terms of precision.

As a classical two-stage object detection method, Faster R-CNN generates candidate regions by a Region Proposal Network (RPN) and relies on a region-based convolutional neural network (R-CNN) for fine-grained classification and bounding box regression for each candidate region. Faster R-CNN shows good accuracy when dealing with larger objects. However, if the object is small (which means the background will be included in the ROI or the noise level is high), Faster R-CNN will always drop a lot of accuracy compared with R-CNN because the information will be lost, and lots of computation is wasted in the process of candidate box generation. However, Faster R-CNN is still used in most high-precision tasks because we can still use the information in region suggestions to guide the box to stay at the target.

In contrast, CSEANet adopts the single-stage detection framework of YOLOv8, retaining its excellent inference speed and real-time performance while incorporating innovative improvements such as the introduction of SAF slicing preprocessing, the optimization of the backbone network with MBConvBlock and CBAM modules, and multi-scale feature fusion through BoltFusionFPN. These advancements significantly enhance the accuracy of small object detection. Experimental results show that CSEANet achieves *mAP*@50 and *mAP*@75 scores of 0.998 and 0.922, respectively, far surpassing Faster R-CNN’s 0.486 and 0.174, as well as YOLOv8’s 0.868 and 0.808. This demonstrates the substantial improvements in both accuracy and small object detection performance provided by CSEANet.

CSEANet’s advantages are particularly prominent in small object detection. By utilizing SAF slicing preprocessing, CSEANet more effectively captures the features of small objects, reducing the interference from background noise and allowing small objects to be more clearly represented and detected. Furthermore, CSEANet employs MBConvBlock and CBAM modules in the backbone network to optimize shallow feature extraction, enabling the network to more precisely focus on small object regions and further enhancing detection performance. Additionally, by replacing the C2f module in YOLOv8 with BSBlock, CSEANet effectively reduces computational redundancy, improves feature extraction efficiency, and mitigates the common computational overhead issues found in Faster R-CNN and YOLOv8.

The success of CSEANet demonstrates the immense potential of single-stage detection frameworks in enhancing small object detection accuracy, especially when combined with lightweight strategies and optimized feature fusion techniques. Through innovative designs in the backbone and neck networks, CSEANet not only significantly outperforms traditional two-stage methods in detection accuracy but also overcomes the limitations of Faster R-CNN and similar methods in small object detection while maintaining high inference speed. Therefore, CSEANet provides an efficient and accurate solution for small object detection tasks in complex backgrounds.

### 4.6. Visualization

To further demonstrate the effectiveness of our proposed method, we present multiple representative detection results in Figure 9, derived from UAV-captured images of railway steel truss bridges. The selected examples reflect a range of inspection scenarios, including different camera angles and bolt regions with varying levels of spatial density—from sparse configurations to areas where bolts are densely packed.

As shown in Figure 9, CSEANet successfully detects and classifies surface bolts across different structural positions. The model accurately identifies bolts in both central and peripheral regions of the image, demonstrating strong spatial generalization. Moreover, all three defect categories—boltNormal, boltCorrosion, and boltMissing—are correctly recognized and clearly annotated with distinct bounding boxes. Even in visually complex areas, the model maintains high localization precision and category-level accuracy.

Each result in Figure 9 corresponds to outputs from the same trained model used in our quantitative evaluations (Table 3, Table 4, Table 5, Table 6, Table 7 and Table 8), ensuring consistency between the visual evidence and reported metrics. These results confirm the model’s robustness and reliability under realistic UAV inspection conditions, offering strong qualitative support for its practical deployment.

## 5. Limitations and Weaknesses

Despite the promising results of CSEANet for detecting surface defects on bolted connections, several limitations must be acknowledged.

A key limitation is the model’s performance under varying environmental conditions, particularly lighting, shooting distances, and camera angles. These factors can significantly impact detection accuracy. While the system performs well under controlled conditions, real-world environments, especially those involving UAV-based inspections, introduce variability that can obscure bolt defects due to changes in lighting or perspectives.

Additionally, background noise in UAV imagery presents a challenge. The limited diversity in the dataset, particularly regarding background types, may reduce the system’s robustness in real-world applications where backgrounds are often more complex. Although the SAF preprocessing method improves performance, more diverse training data is necessary to enhance the model’s generalizability.

Furthermore, UAVs, while efficient, face challenges like camera calibration, flight stability, and image distortion, especially in hard-to-reach areas. These issues could affect the quality of imagery and, in turn, detection accuracy.

Finally, comparing this study with works like those of Jafari et al. (2025) [61] highlights that accounting for diverse bolt shapes and backgrounds remains a challenge. While our method is robust, incorporating more varied datasets would improve its applicability to a wider range of real-world conditions [62].

## 6. Conlusions

This paper addresses the challenge of detecting surface bolt defects in railway steel truss bridges using UAV-based vision systems. To improve detection accuracy and efficiency, we propose CSEANet, an enhanced YOLOv8-based framework that integrates SAF slicing for improved small object representation and structural optimizations—MBConvBlock, BSBlock, and BoltFusionFPN—for better feature extraction and multi-scale fusion.

Experimental results show that CSEANet achieves high performance (*mAP*@50:95 of 0.952) and real-time capability, outperforming baseline methods, particularly in small object scenarios. These results suggest strong potential for UAV-based deployment in real-world inspection tasks.

However, the current dataset is relatively small (53 original images before augmentation), which may limit the generalizability of the results. Although SAF preprocessing expands the training set and improves model robustness, future work will focus on collecting a larger, more diverse dataset and providing visual examples to better demonstrate data variability and detection effectiveness.

In addition, future research will explore adapting the method to detect other structural defects and deploying the model on embedded UAV platforms with limited computational resources, to further enhance its practical applicability in infrastructure inspection.

## Figures and Tables

**Figure 1 sensors-25-03500-f001:**
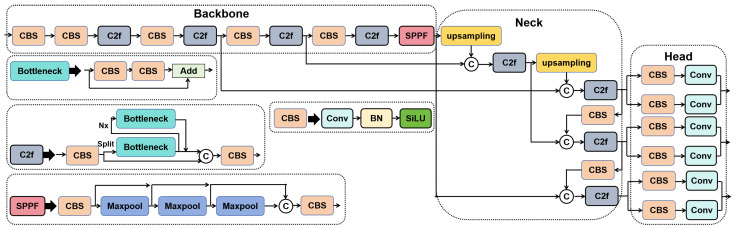
The structure of YOLOV8.

**Figure 2 sensors-25-03500-f002:**
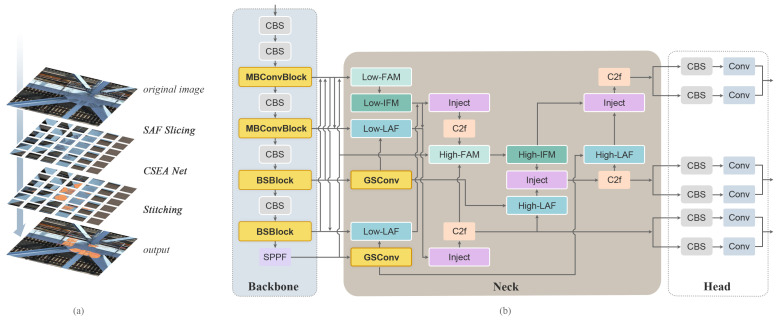
Figure (**a**) shows the process image of surface bolt defect detection for railway steel truss bridges, and Figure (**b**) illustrates the structure of CSEANet.

**Figure 3 sensors-25-03500-f003:**
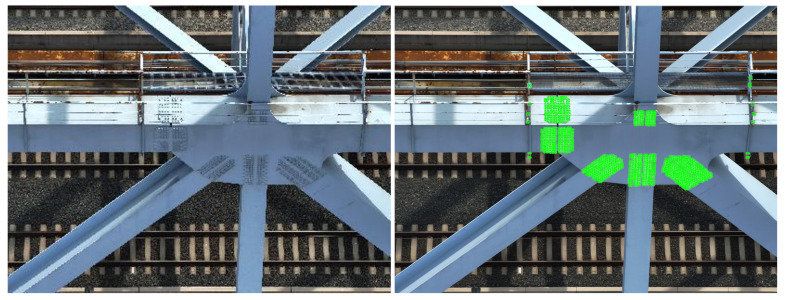
Schematic diagram of railway steel truss bridge collected during UAV patrol inspection. The part marked in green is a bolt.

**Figure 4 sensors-25-03500-f004:**
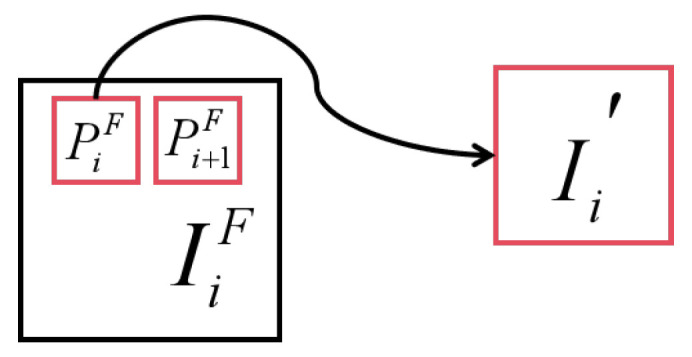
Schematic diagram of SAF process.

**Figure 5 sensors-25-03500-f005:**
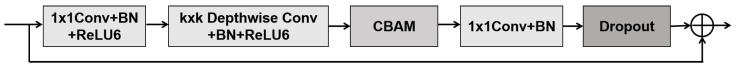
The structure of MBConvBlock.

**Figure 6 sensors-25-03500-f006:**
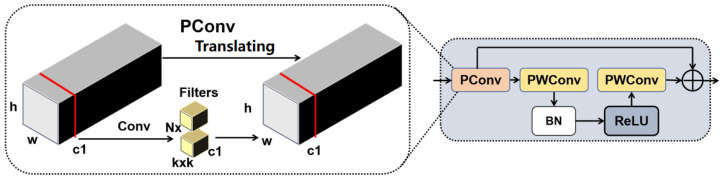
The structure of BSBlock.

**Figure 7 sensors-25-03500-f007:**
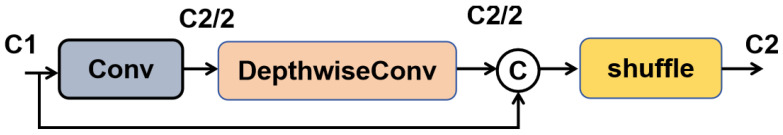
The structure of GSConv.

**Figure 8 sensors-25-03500-f008:**
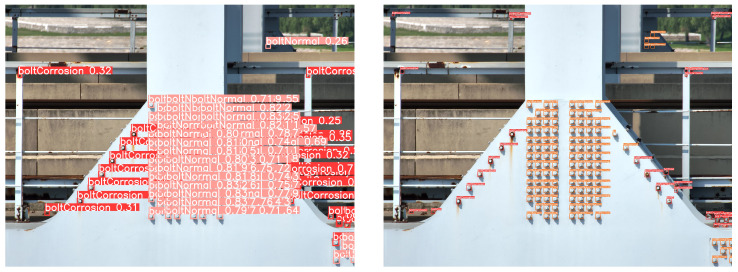
Comparison of algorithm results (**left**) without, (**Right**) with SAF preprocessing.

**Figure 9 sensors-25-03500-f009:**
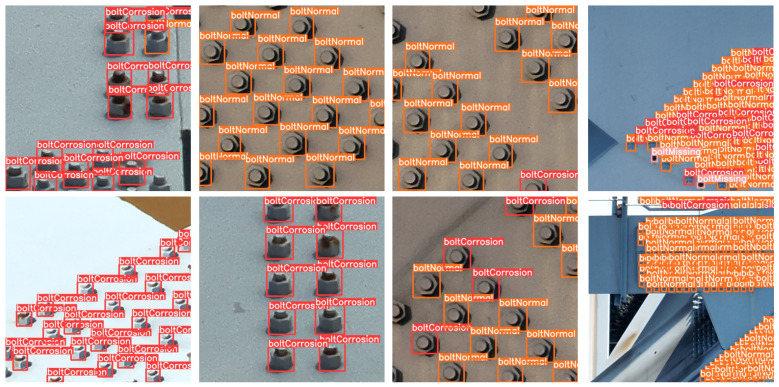
Final results of surface bolts defect detection for the railway steel truss bridge. The image demonstrates the detection of various defects, including bolt corrosion and missing bolts across different sections of the bridge.

**Table 1 sensors-25-03500-t001:** Summary of dataset composition after SAF slicing.

Dataset Split	Number of Images	Image Size (Pixels)
Training Set	891	640 × 640
Validation Set	112	640 × 640
Test Set	112	640 × 640

**Table 2 sensors-25-03500-t002:** Training parameters.

Parameter	Value
Epochs	300
Batch size	16
Learning rate	0.0001
Weight decay	$5\times 10^{-4}$

**Table 3 sensors-25-03500-t003:** Comparative experimental results of SAF preprocessing.

YOLOv8s	SAF	*mAP*			
		**All**	**boltCorrosion**	**boltMissing**	**boltNormal**
✓		0.414	0.479	0.254	0.510
✓	✓	0.818	0.768	0.769	0.916

**Table 4 sensors-25-03500-t004:** Experimental results of model ablation with all labels.

YOLOv8s	MBConvBlock	BSBlock	BoltFusionFPN	*P*	*R*	*mAP*
✓				0.828	0.807	0.818
✓	✓			0.848	0.806	0.871
✓		✓		0.886	0.807	0.837
✓			✓	0.872	0.793	0.846
✓	✓	✓	✓	0.921	0.866	0.877
✓	✓	✓	✓	0.9	0.945	0.952

**Table 5 sensors-25-03500-t005:** Experimental results of model ablation with the boltCorrosion label.

YOLOv8s	MBConvBlock	BSBlock	BoltFusionFPN	*P*	*R*	*mAP*
✓				0.731	0.779	0.768
✓	✓			0.718	0.772	0.791
✓		✓		0.765	0.788	0.811
✓			✓	0.735	0.742	0.785
✓	✓	✓	✓	0.826	0.838	0.857
✓	✓	✓	✓	0.828	0.912	0.916

**Table 6 sensors-25-03500-t006:** Experimental results of model ablation with the boltMissing label.

YOLOv8s	MBConvBlock	BSBlock	BoltFusionFPN	*P*	*R*	*mAP*
✓				0.829	0.75	0.769
✓	✓			0.898	0.75	0.888
✓		✓		1	0.693	0.771
✓			✓	0.895	0.75	0.845
✓	✓	✓	✓	0.95	0.784	0.8
✓	✓	✓	✓	1	0.973	0.995

**Table 7 sensors-25-03500-t007:** Experimental results of model ablation with the boltNormal label.

YOLOv8s	MBConvBlock	BSBlock	BoltFusionFPN	*P*	*R*	*mAP*
✓				0.923	0.892	0.916
✓	✓			0.929	0.897	0.935
✓		✓		0.894	0.942	0.931
✓			✓	0.908	0.888	0.907
✓	✓	✓	✓	0.958	0.977	0.974
✓	✓	✓	✓	0.872	0.951	0.945

**Table 8 sensors-25-03500-t008:** Performance comparison of different methods.

Method	Backbone	*mAP*@50	*mAP*@75	*mAP*@50:95	FLOPs (G)	Params (MB)
Faster-rcnn	ResNet50	0.486	0.174	0.228	90.908 G	41.358
cascade-rcnn	ResNet50	0.487	0.168	0.231	0.119 T	69.158
RetinaNet	ResNet50	0.259	0.08	0.12	81.92 G	36.371
CenterNet	ResNet50	0.749	0.325	0.35	80.495 G	32.116
YOLOv8s	–	0.868	0.808	0.818	28.6 G	11.2
CSEANet (ours)	–	0.998	0.922	0.952	28.7 G	44.55

## Data Availability

The data presented in this study are not publicly available due to privacy and security considerations related to critical infrastructure information. Requests for data access may be considered on a case-by-case basis by contacting the corresponding authors.

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
