# Peer review of "CSEANet: Cross-Stage Enhanced Aggregation Network for Detecting Surface Bolt Defects in Railway Steel Truss Bridges"

_sensors, 2025, doi:10.3390/s25113500_

Round 1
Reviewer 1 Report
Comments and Suggestions for Authors
Manuscript Title: CSEANet: Cross-Stage Enhanced Aggregation Network for Detecting Surface Bolt Defects in Railway Steel Truss Bridges
Strengths
The manuscript addresses a highly practical and timely issue: the detection of bolt defects in steel truss bridges, which is critical for ensuring structural safety. The use of UAV-based imagery is appropriate for this type of inspection and shows promise in terms of scalability.
The proposed model, CSEANet, builds upon YOLOv8 and incorporates several noteworthy modifications, including SAF slicing, MBConv blocks with attention mechanisms (CBAM), and a module referred to as BoltFusionFPN for improved multi-scale feature extraction. These additions appear to be well motivated.
The experimental section is solid and includes a variety of comparisons with baseline methods as well as some ablation studies. It is commendable that the authors went beyond simply demonstrating that “it works.”
The model performs particularly well on small object detection tasks, which are especially challenging in UAV-based imaging scenarios.
Weaknesses
The dataset used is quite limited in scope (only 53 images), which raises valid concerns regarding generalisation and potential overfitting. It is unclear to what extent slicing improves the dataset diversity or mitigates the lack of training examples.
Several components of the methodology are insufficiently explained — for example, the practical implementation of SAF slicing or the specific structure of the BoltFusionFPN module. Including schematic diagrams would be helpful in clarifying these aspects.
There is no detailed error analysis provided. It would be useful to know which types of misclassifications are most common (e.g., confusion between defect classes, or high false positive rates).
The manuscript does not specify whether the dataset or source code will be made publicly available, which is increasingly expected for reproducibility in deep learning research.
From what can be inferred, all data appear to originate from a single site. No cross-site testing is reported, which limits the assessment of the model’s generalisability.
Suggestions for Improvement
Methodology
• Consider adding a block diagram illustrating the architecture of CSEANet. It is difficult to conceptualise the full design based solely on the text. A side-by-side comparison with standard YOLOv8 would be especially useful.
• Clarify how overlapping slices are processed during inference. Is any post-processing (e.g., non-maximum suppression across slices) applied?
• Explicitly state whether slicing is applied during both training and inference, and whether it is performed per image or per class.
Dataset and Evaluation
• Include a breakdown of the number of samples per class (boltCorrosion, boltMissing, boltNormal) to clarify dataset balance.
• Provide additional information on the annotation process — for instance, how many annotators were involved and whether any inter-annotator agreement measures were used. This would strengthen the dataset’s credibility.
• Ideally, evaluate the model on data from different bridges or under different environmental conditions to assess its ability to generalise.
Results and Analysis
• Include some form of error analysis — for example, confusion matrices, visualisation of failure cases, or quantification of false positives and negatives.
• Precision–recall curves or per-class F1-scores would complement the reported mAP and offer more insight into class-wise performance.
Transparency and Ethical Considerations
• Indicate whether the dataset and code will be made available. If not, briefly explain the reason (e.g., data sensitivity, project confidentiality, etc.).
• Discuss practical limitations, such as dependencies on UAV flight parameters (altitude, angle) and the challenges of real-time deployment.
Conclusions
• Consider expanding the conclusions to include a brief discussion of the current limitations of the method and potential future directions — for example, adaptation to other defect types or deployment on embedded UAV platforms.
Recommendation: Minor to Moderate Revision
The paper represents a useful and timely contribution to UAV-based bridge inspection and computer vision applications. The concept is promising and the results are encouraging, but there are several areas — particularly regarding methodological clarity, data transparency, and reproducibility — that should be addressed before the manuscript can be considered for publication.
Language and Style Comments:
The manuscript is generally well written and easy to follow in terms of technical content. Its structure is clear, and I had no difficulty understanding the core ideas. That said, I believe the English could be polished to enhance overall readability. While not a major issue, the language occasionally feels a bit stiff or repetitive.
Here are a few points that stood out:
• Repetitive phrasing – Phrases such as "this paper proposes", "we introduce", and "in this study" appear quite frequently, sometimes multiple times within a short span. Varying the way ideas are introduced could improve the text’s flow and make it more engaging.
• Long or densely packed sentences – In several places, the authors attempt to fit too much information into a single sentence. Breaking some of these up would make the paper easier to read, especially for non-specialist readers.
• Awkward phrasing – Some expressions feel slightly unnatural or overly formal. For example, instead of "formulate the corresponding plan for the maintenance", something like "develop a maintenance plan" would be clearer and more idiomatic. Similarly, "bolt falling off" might be more appropriately expressed as "bolt detachment" or "loss of a bolt."
• Terminology inconsistency – There appears to be some inconsistency in terminology; for instance, both "slice-assisted fine-tuning" and "slicing-aided fine-tuning" are used. While minor, maintaining consistent phrasing can help avoid confusion.
• Ambiguous formulations – A few descriptions, particularly in the methods section, are somewhat difficult to interpret due to phrasing. Terms like "reconstruction of overlapping slices" or "reducing redundancy" would benefit from clearer, more concrete explanations — possibly even a brief example to illustrate what is meant.
In summary, I recommend that the manuscript undergo a language review by a native English speaker or a professional editor experienced in academic writing. The technical content is strong, and with some refinement, the writing will reflect that strength more effectively.
Reviewer 2 Report
Comments and Suggestions for Authors The approach proposed by the authors appears promising and may have practical applications, particularly for the inspection of railway infrastructure and other domains where detection of bolt connection defects is required. However, the reviewed manuscript raises concerns regarding the correctness of the experimental procedure and the reliability of the presented results. In light of these issues, the work requires substantial revision, and the reported outcomes should be subject to further verification. Although not mandatory, publishing the source code of the proposed algorithm would be a commendable step toward scientific transparency and would enable independent validation by other researchers, helping to dispel any potential doubts.- The abstract mentions an accuracy improvement of 97.1% mAP. However, an analysis of Table 6 indicates that the improvement between the baseline YOLOv8s model and the proposed modification amounts to a maximum of 16.4%. While this is a positive result, it does not correspond to the figure stated in the abstract. It is recommended that the abstract be revised to accurately reflect the actual results.
- The dataset, which consists of only 53 images, is too small to draw substantiated conclusions about the advantages of the proposed method, especially given the high risk of overfitting. Since no image examples are provided by the authors, it is not possible to assess the dataset's diversity. It is recommended to expand the dataset and include representative samples from both the training and validation sets in the manuscript.
- The technical specifications and hyperparameters of the model should preferably be presented in the form of a table to improve readability and ensure transparency.
- The manuscript lacks training progress graphs or descriptions of early stopping mechanisms, which makes it difficult to assess potential overfitting. It is recommended to include such information.
- The hyperparameters of the models compared in Table 6 are not provided in the text. It is advisable to compile the parameters of all compared models into a single table. Moreover, considering the limited dataset size (53 images), it would be useful to discuss whether using larger models under such conditions is appropriate.
- An analysis of Figure 8 reveals that the model failed to detect a significant portion of the bolts, which contradicts the high accuracy metrics presented in Tables 2–6. It is recommended to address these inconsistencies and, if possible, provide additional examples.
- The manuscript lacks a discussion section in which the limitations of the proposed approach could be outlined. It is recommended to add such a section.
- The conclusions should be revised after the dataset has been expanded and the experiments repeated.
Reviewer 3 Report
Comments and Suggestions for Authors
This paper presents a novel deep learning architecture, CSEANet (Cross-Stage Enhanced Aggregation Network), designed for detecting surface bolt defects in railway steel truss bridges using image data. The authors propose a multi-stage feature aggregation mechanism th to enhance defect recognition accuracy. Overall, the paper makes a meaningful contribution to the field of intelligent inspection and structural monitoring using UAV-based imaging and computer vision techniques. I recommend a major revision to improve technical clarity and reproducibility.
Please revise the manuscript to provide a detailed comparison of the advantages and disadvantages of your proposed approach with previous work, specifically:
- Bolt Connection Assessment: Compare your method with the following studies that utilize UAV imagery and deep learning for bolted connection inspection and crack detection:
- Jafari, F., & Dorafshan, S. (2025). Condition assessment of bolted connections in steel structures using deep learning. Innovative Infrastructure Solutions, 10(2), 65.
- Jafari, F., Dorafshan, S., & Kaabouch, N. (2023, June). Segmentation of fatigue cracks in ancillary steel structures using deep learning convolutional neural networks. In 2023 IEEE/ASME International Conference on Advanced Intelligent Mechatronics (AIM) (pp. 872–877). IEEE.
- Figure 1: Provide more information and discussion related to Figure 1. Explain what the image demonstrates, including the dataset source, annotations, and any preprocessing involved.
- Equations:
- Number all equations consistently throughout the manuscript (e.g., Eq. (1), Eq. (2), etc.).
- Refer to each equation explicitly in the text and describe the significance of each term.
- Provide appropriate citations for the algorithms and formulas used.
- Training and Dataset Details:
- Include the total training time, hardware specifications (e.g., GPU used), and the number of epochs.
- Report the number of bolts in the dataset, categorizing them as sound or defective samples.
- Background Influence on Algorithm Performance:
- Discuss how varying the background (e.g., sky, vegetation, concrete, steel, shadowed regions) in UAV images may affect model performance.
- If available, provide evidence from experiments using different backgrounds and report corresponding performance metrics (e.g., IOU, accuracy, precision, recall, F1-score).
- Figure 8:
- Modify Figure 8 to include subfigures labeled as (a), (b), etc., and describe each subfigure clearly in the caption and body text.
- Sensor Specifications:
- Provide information on the UAV sensor specifications, including:
- Image resolution (in pixels),
- Focal length of the camera (in mm),
- Flight altitude, if it affects spatial resolution.
Make sure all acronyms were introduced in paper for first time like name of algorithms, UAV ,,………….
Reviewer 4 Report
Comments and Suggestions for Authors
The authors investigated a novel bolt defect detection method via computer vision algorithms. The method has merits in a few aspects, such as computational speed and accuracy. However, the reviewer has some comments below that can improve the quality of the paper:
- The size of the bolt detections is small, with only 53 images. Although after sliding, it offers more image patches, the reviewer wondered how the proposed method works in other datasets, especially the bridges that have different surface layouts or bolt configurations.
- During the literature review, the authors mentioned that many existing methods have a high computational cost and may not be in real-time (off-board). Does the method proposed in this study fully address these limitations? It seems like no information is reported.
- The contributions claimed by this study, shown on page 3, are some incremental technological improvements, not true contributions that can bridge the research gap. The authors shall revisit Sections 1 and 2, discuss how their proposed method can help peers to advance the field.
- What are the criteria for labeling missing bolts and corrosion bolts? Structurally, how to define bolt corrosion? Are image patches of missing bolts and bolts with corrosion part of the normal bolt dataset?
- Did the authors investigate the proposed method under different lighting conditions, shooting distances, and camera angles against the bridge surface? These factors seem to negatively affect the performance of the proposed method.
Reviewer 5 Report
Comments and Suggestions for Authors
- “In the pictures acquired by UAVs, the bolts are very small targets, only taking up a few pixels in high-resolution pictures, which makes it hard for traditional object detection algorithms to identify them.” UAV applications for bolt loosened or defects were done through for the first time: Real-time multiple damage mapping using autonomous UAV and deep faster region-based neural networks for GPS-denied structures. This should be discussed.
- Before discussing of deep learning for loosened bolt detection, there are computer vision based approaches: Vision-based detection of loosened bolts using the Hough transform and support vector machines; Fully automated vision-based loosened bolt detection using the Viola–Jones algorithm. These representative works should be discussed.
- “UAVs are usually equipped with devices that have limited computational power and high-precision object detection algorithms require a large amount of computational resources during inference. In practical applications, detection algorithms should focus on not only high accuracy but also lightweight design and real-time performance to meet the requirements of UAVs applied in inspection tasks.” The most critical problems of the UAV applications are the manual control of the UAV that requires at least licensed skilled two pilots who are very expensive. Therefore autonomous UAVs were developed for the first time in this area: Autonomous UAVs for structural health monitoring using deep learning and an ultrasonic beacon system with geo‐tagging; This should be discussed.
- The next limitations are the unexpected obstacles for the flight of UAVs. To overcome this technical challenges, autonomous UAVs with obstacle avoidance was also developed for the first time: Deep learning-based obstacle-avoiding autonomous UAVs with fiducial marker-based localization for structural health monitoring. This original paper should be discussed.
- The first application of CNN for the bolt damage detection was: Autonomous structural visual inspection using region‐based deep learning for detecting multiple damage types. This should be discussed.
- What is new technical contributions of this paper: YOLOv8 is an existing network that you can download and train it with data.
- This is clearly the topic of deep learning based SHM therefore the key and original papers should be discussed: Deep learning-based crack damage detection using convolutional neural networks; Deep learning based structural health monitoring.
- All the data used for training and testing should be tabulated in terms of number of images and their sizes.
Round 2
Reviewer 2 Report
Comments and Suggestions for Authors
I recommended publishing the article in present form
Author Response
I would like to sincerely thank you for your careful review and valuable comments on the manuscript. I am particularly grateful for your recommendation to publish the article in its present form. Your positive feedback is truly encouraging.
Reviewer 3 Report
Comments and Suggestions for Authors
Thank you for your detailed response to my comments. I kindly ask that you double-check the references to ensure that all relevant citations are included in the final manuscript.
Based on my previous search related to background removal and bolt detection, the paper I mentioned appears to be the only study that discusses the diversity of bolt shapes and varying backgrounds. However, I could not find it cited in the revised version dated May 23.
Including all relevant prior studies in the literature review would be valuable, as future investigations may refer to earlier works through your paper. Even though your approach improves upon previous methods and uses a different dataset, acknowledging related studies provides important context and continuity for future research.
To summarize, I was unable to locate the corresponding citations in the revised manuscript. Please review the references to ensure they are properly cited and ordered, if you agree with their inclusion based on your response. I am suggesting this based on my earlier search, as I did not find other studies focusing on both bolt detection and background removal with diverse bolt shape.
Author Response
Thank you so much for your thoughtful and detailed feedback. We deeply appreciate your continued support in improving our manuscript.
In response to your request, we have thoroughly reviewed the references to ensure that all relevant citations, including the ones you mentioned, are properly included in the final manuscript. We have now added the two important studies that you referred to:
-
Jafari, F., & Dorafshan, S. (2025). Condition assessment of bolted connections in steel structures using deep learning. Innovative Infrastructure Solutions, 10(2), 65.
-
Jafari, F., Dorafshan, S., & Kaabouch, N. (2023, June). Segmentation of fatigue cracks in ancillary steel structures using deep learning convolutional neural networks. In 2023 IEEE/ASME International Conference on Advanced Intelligent Mechatronics (AIM), pp. 872–877. IEEE.
These studies are now correctly cited in the manuscript as references [62] and [63], respectively. We believe that including these references not only addresses your valuable comment but also strengthens the continuity and context for future investigations in the field.
We have carefully reviewed the entire manuscript to ensure that all citations are appropriately included, and we hope that these revisions now meet your expectations.
Once again, thank you for your insightful suggestions, which have significantly enhanced the quality of our work. We truly appreciate your time and effort in reviewing our manuscript, and we are grateful for your constructive input.
Please feel free to let us know if you have any further comments or if there’s anything else we can address.
Reviewer 4 Report
Comments and Suggestions for Authors
I recommend this paper for publication if the author can address the comments below:
Please mention the limitations of this study in addressing lighting conditions, shooting distances and camera angles against the bridge surface.
Author Response
Comments 1: Please mention the limitations of this study in addressing lighting conditions, shooting distances and camera angles against the bridge surface.
Response 1:
Thank you so much for your valuable feedback and thoughtful suggestions. We truly appreciate your careful review of our manuscript, and your comments have helped to improve the quality of our work significantly.
In response to your request to address the limitations of our study regarding lighting conditions, shooting distances, and camera angles against the bridge surface, we have added a detailed discussion of these aspects in the "Limitation and Weakness" section. Specifically, we have highlighted how varying lighting conditions, shooting distances, and camera angles can impact the model's performance. We also acknowledged the challenges these factors present in real-world UAV-based inspections and discussed the need for future work to consider more diverse environmental variables.
To make it easier for you to spot the revisions, we have highlighted the newly added sections in yellow in the revised manuscript. We believe this addresses your concern effectively and provides a clearer understanding of the challenges faced by our method in practical applications.
Once again, we sincerely thank you for your insightful feedback. Your input has been invaluable in refining our work, and we hope the revisions meet your expectations. Please let us know if there are any further aspects you'd like us to address.
Reviewer 5 Report
Comments and Suggestions for Authors
All the comments were addressed.
Author Response
我衷心感谢您的仔细审阅和对手稿的宝贵评论。我特别感谢您的建议,以目前的形式发布这篇文章。您的积极反馈确实令人鼓舞。